# Tef (*Eragrostis tef*) Responses to Phosphorus and Potassium Fertigation under Semi-Arid Mediterranean Climate

**Moshe Halpern [1,†], Kelem Gashu [1,2,†], Isaac Zipori [1], Yehoshua Saranga [2] and Uri Yermiyahu [1,2,*]**

[1] Gilat Research Center, Agricultural Research Organization, Negev 85280, Israel; moshehalpern416@hotmail.com (M.H.); kelem-ga.alamrie@mail.huji.ac.il (K.G.); matabsor@volcani.agri.gov.il (I.Z.)

[2] The Robert H. Smith Faculty of Agriculture, Food and Environment, The Hebrew University of Jerusalem, Rehovot 76100, Israel; shuki.saranga@mail.huji.ac.il

[*] Correspondence: uri4@volcani.agri.gov.il

[†] Equal contribution.

**Abstract:** Tef (*Eragrostis tef* (*Zucc.*) Trotter) is an annual small grain, panicle bearing, C4 cereal crop native to Ethiopia, where it is a major staple food. The objectives of the present study were to characterize the responses of two tef genotypes to escalating phosphorus (P) and potassium (K) levels and to determine an optimum range for P and K at which tef performance is maximized. Two experiments were carried out in the Gilat Research Station, each testing two different genotypes of tef (405B and 406W), one experiment in pots in controlled conditions, and the other in the field. In both experiments, the highest grain yield increased until 6 mg L$^{-1}$ P, and declined at 12 mg L$^{-1}$ P. The decline was precipitous and significant in the pot experiment, and gradual and statistically insignificant in the field experiment. In the pots experiment, the grain yield increased until 40 mg L$^{-1}$ K, with no significant decrease thereafter. The effect of K concentration was only seen in the grain yield and not in the size of the other plant organs. In the field experiment, grain yield was highest at 80 mg L$^{-1}$ K, but it was not statistically different from 40 mg L$^{-1}$. The effect of K on growth was only apparent at maturity and not at flowering.

**Keywords:** tef; fertigation; fertilizer optimization

## 1. Introduction

Tef (also written as "teff") (*Eragrostis tef* (*Zucc.*) Trotter) is an annual small grain, panicle bearing, C4 cereal crop native to Ethiopia, where it is a major staple food, accounting for two thirds of the daily protein intake of the population, and 11% of the caloric intake [1]. It is resistant to many biotic and abiotic stresses, making it an attractive crop for smallholder farmers [1]. It is known for its many health benefits, including its low glycemic index and high concentrations of essential amino acids, and it is also gluten free [2].

The majority of tef cultivation is located in Ethiopia [1], and it is mostly cultivated under low input, rainfed conditions However, in recent years interest has increased in cultivating tef outside of its native environment [1], including drylands where irrigation is required. Interest has also increased in cultivating tef in a more intensive manner [1,3]. One way of intensifying tef cultivation would be to apply water and fertilizer through irrigation (fertigation), which could allow for a high yield even in conditions of low rainfall or low soil fertility. Applying nutrients through fertigation requires a detailed understanding of how the crop responds to different concentration of nutrients, of which nitrogen (N), phosphorus (P) and potassium (K) are the most prominent. The response of tef to N fertilizer has been described in a previous paper [4], and in this paper we will discuss the response of tef to P and K.

Phosphorus fertilization of tef is common in Ethiopia [5]. A blanket recommendation by the Ethiopian Ministry of Agriculture was to apply between 30 and 40 kg P ha$^{-1}$, depending on the soil type [5], which was later changed to 100 kg DAP ha$^{-1}$ regardless of the soil type. In recent years, soil tests have been used to improve accuracy of P fertilization [5]. A number of field trials have been published in recent years showing the effect of different levels of P on tef growth parameters. Dubale [6] conducted a field experiment in the North Shewa Highlands of Ethiopia with three different rates of P fertilizer (0, 50, and 100 kg P$_2$O$_5$ ha$^{-1}$) and found that broadcasting different amounts of P fertilizer at sowing did not significantly affect P concentration in plant tissue or grain yield, perhaps because there was already sufficient P in the soil in which the experiment was conducted or the P that was applied became unavailable to plants. Dereje et al. [7] conducted a full factorial field experiment in the Assosa Zone in Ethiopia in which five levels of N and four levels of P were tested. They found significant main effects and interactive effects of N and P on many different growth parameters including yield. After economic analysis, they concluded that in soils similar to their test plots, the best practice would be to apply 10 Kg P ha$^{-1}$, along with 46 Kg N ha$^{-1}$.

Although K is universally recognized as important for crop growth and yield, for many years tef grown in Ethiopia was usually fertilized only with N and P and not with K [8]. This was due to the common belief, stemming from studies conducted in the 1960s and 1970s, that claimed that K was not limiting in Ethiopian agricultural soils [8,9]. Recently, this idea was called into question with largescale mapping of Ethiopian soils in a project initiated by the Ethiopian government, called the Ethiopian Soil Information System (Ethiosis) [10]. A number of field trials have been recently published indicating that K application can increase tef yield [9,11]. Similarly, a greenhouse trial in which different levels of K were applied to tef growing in pots with different Ethiopian soils was conducted in order to determine an optimum K fertilization rate [12].

The purpose of this study is to elucidate the response of tef to different levels of P and K applied in the irrigation water. It was carried out in two phases. In the first phase, tef was grown under controlled conditions in a greenhouse, in perlite, an inert medium that allowed full control of the nutrient concentration in the root zone. This is in contrast to soil, which can absorb, fix, or release ions of K$^+$ and PO$_4^-$, making it difficult to determine the exact nutrient concentrations to which the roots are exposed. The second phase involved growing tef out in the field in a semi-arid Mediterranean climate in natural soil. While the first phase was designed to clarify the basic response of tef to different P and K concentrations, the second phase was intended to show how the lessons learned from the first phase play out in a realistic agricultural environment. The same experiments were used to study N fertigation in tef, recently published by Gashu et al. [4].

## 2. Materials and Methods

### 2.1. Plant Materials and Experimental Design

A pot experiment and a field experiment were carried out in the winter of 2015–2016 and the summer of 2016, respectively, in order to determine the response of tef to N, P and K nutrition. These experiments contained different segments in which different nutrients were varied, but which shared a control treatment. The N segment of these experiments has already been reported on by Gashu et al. [4]. The P and K segments of the experiments were conducted at the same time and under the same experimental conditions as the published N segment.

Two tef genotypes from the Israel Gene Bank (Agricultural Research Organization, Volcani Center, Rishon Lezion, Israel) were used in the current study, RTC-405 and RTC 406 (denoted hereafter as 405B and 406W, with the last letter indicating brown or white seed color). The entire collection was propagated and initially phenotyped in a common garden experiment during spring 2015 [13].

The pot experiment was conducted in a walk-in plastic-covered tunnel (6 m-wide, 2.4 m tall and 30 m-long) at the Gilat Research Center, Negev, Israel during winter 2015–2016. The P segment of the experiment comprised 4 treatments of P concentrations in the irrigation solution (1, 3, 6, and 12 mg $L^{-1}$ P) and the K segment of the experiment comprised 4 treatments of K concentration in the irrigation solution (10, 20, 40, 80 mg $L^{-1}$ K). Both segments included two tef genotypes (405B and 406W). A factorial (P or K treatments X genotypes) randomized block design was employed with five replicates. The P and K segments of the experiment shared a control group (N, P and K = 40, 6, and 40 mg $L^{-1}$, respectively). The concentration of the other minerals in the fertigation solution of all treatments were: 40 mg $L^{-1}$ N, 20 mg $L^{-1}$ Ca, 20 mg $L^{-1}$ Mg, 28 mg $L^{-1}$ S, 0.3 mg $L^{-1}$ B, 0.6 mg $L^{-1}$ Fe, 0.3 mg $L^{-1}$ Mn, 0.15 mg $L^{-1}$ Zn, 0.02 mg $L^{-1}$ Cu, and 0.02 mg $L^{-1}$ Mo. No 0 P or 0 K treatments were included, since the perlite is an inert medium and the plants simply cannot grow without at least a small dose of P and K. More details regarding the environmental conditions and a timeline of the experiment are described in Gashu et al. [4]. A full list of treatments and replications, including those reported on by Gashu et al. [4], are listed in Appendix A.

The field experiment was conducted at the Gilat Research Center during the summer of 2016. The P segment of the experiment comprised 4 treatments of P concentrations in the irrigation solution (0, 3, 6, and 12 mg $L^{-1}$ P) and the K segment of the experiment comprised 3 treatments of K concentration in the irrigation solution (0, 40 and 80 mg $L^{-1}$ K). Both segments included two tef genotypes (405B and 406W). A factorial (P or K treatments × genotypes) split-plot block design with 5 replications was used, with fertigation treatment in the main plots and genotypes in subplots. Each main plot (5 m long × 4.2 m wide) consisted of 28 rows (14 rows per genotype) with 15 cm of space between rows. Each main plot was irrigated by 14 drip lines between each pair of rows. Seeds were directly sown on 13 July 2016 into well-prepared dry soil at a depth of ~1 cm, and seeding rate of 800 mg/m², using a hand-driven precision garden seeder (1001B, Earthway, Bristol, IN, USA). During the first two weeks, plots were irrigated to drainage in order to avoid salt accumulation. Two weeks after sowing, fertigation treatments were started by injecting 1L of custom-made fertilizer solutions into 100 L of water. Fertigation was applied daily via a drip system with water amounts determined according to Penman Monteith potential evapotranspiration [13] multiplied by the following crop coefficients: 1.0 for the first 26 days after sowing, 1.2 for 26–30 days, 1.0 for 30–40 days, 1.2 for 40–60 days, 1.0 for 60–70 days and 0.8 from 70 days after sowing to the final harvest as described by Yihun et al. [14]. Both P and K segments shared a control group (N, P and K = 60, 6, 40 mg $L^{-1}$, respectively). A full list of treatments and replications, including those reported on by Gashu et al. [4], are listed in Appendix A. Environmental conditions and a timeline of the experiment are described by Gashu et al. [4].

*2.2. Data Collection*

In both experiments, days to 50% panicle emergence was recorded for each pot and plot by visual observation. The youngest fully expanded leaves from representative plants were selected and used for indirect chlorophyll measurement using SPAD 502 (Minolta Corporation, Ramsey, NJ, USA) chlorophyll meter (85 and 50 days after emergence for pot and field experiment, respectively). In both experiments, plants were sampled twice: at flowering stage (50 and 40 days after sowing for pot and field experiment, respectively) and at maturity stage (final harvest), at 105 (405B) and 114 (406W) days after sowing in the pot experiment, and at 84 and 99 days after sowing (405B and 406W genotypes, respectively) in the field experiment. In the pot experiment, two repetitions, which included 50 pots (25 pots for each sampling time), were sampled by destructive harvest, whereas in the field experiment, tef plants were sampled from 1 × 1 m measured area in each subplot. Sampling was carried out for both genotypes at the same time.

The number of tillers per plant was recorded at maturity. The height and panicle length of the plant was measured at the end of the growing season. In the pot experiment,

the roots were separated from shoots and any loose perlite was washed off with tap water and placed into a paper bag. The shoots were then rinsed several times in distilled water to avoid any contaminants and placed in a different paper bag. Both plant parts were oven-dried for 72 h at 70 °C and weighed to determine the dry matter (DM). Shoots were subsequently used for nutrient analysis.

Lodging was evaluated visually at the end of the winter pot experiment using the following scale: <20% lodged = 1, 20–40% lodged = 2, 40–60% lodged = 3, 60–80% lodged = 4, 80–100% lodged = 5. In the summer field experiment, all the plots were fully lodged by the end of the experiment and, therefore, were not evaluated.

Once harvested, the grains were separated from the straw by hand (pot experiment) or threshing machine (field experiment). The dried tef shoots were then ground in a grinding machine. Approximately 0.1 g of the ground samples and grain was digested by 2 mL sulfuric acid under 180 °C and supplemented with hydrogen peroxide [15]. The concentration of P was determined by an autoanalyzer (Lachat Instruments, Milwaukee, WI, USA) and the concentration of K was determined using an atomic absorption flame photometer (Corning 400, Corning, New York, NY, USA). Mineral uptake of shoot and grain was calculated by multiplying each mineral concentration with respective vegetative DM and grain yield.

Before treatments were applied in the field trial, soil samples from three layers (0–30 cm, 30–60 cm, and 60–90 cm-depth) were collected at five locations within the experimental plot. Soil samples were then oven-dried for 96 h at 70 °C and ground to pass through a 2 mm sieve. Soil analysis included: pH, electrical conductivity (EC), moisture content, K, P, $NO_3$, Mg, Na and Cl in a saturated paste. Phosphorus was extracted using the Olsen bicarbonate method [16], and measured using an autoanalyzer (Lachat Instruments, Milwaukee, WI, USA). Potassium was extracted using ammonium acetate and measured using an atomic absorption flame photometer (Corning 400, Corning, New York, NY, USA). Average values of these soil properties at each depth were previously published [4]. The highest concentration of P and K was found in the 0–30 cm soil strata, at 1.1 mg/kg and 38.8 mg/kg, respectively.

*2.3. Statistical Analyses*

For each set of nutrient treatments (P or K levels), a two-way analysis of variance (ANOVA) was carried out for each variable to determine the effects of genotype, nutrient (P or K), and genotype X nutrient interaction using JMP 13.0 (JMP, Cary, NC, USA) software. Mean separations were performed by the Tukey–Kramer honest significant difference (HSD) test at $p = 0.05$.

## 3. Results

*3.1. Effects of P and K on Different Genotypes*

P and K mostly had the same effects on growth and nutrient content, regardless of the genotype. This is reflected by the fact that the interactive effect (nutrient X genotype) effect was rarely significant, with the exception of the K X genotype effect on yield in the pot experiment ($p = 0.0046$) (Figure 1F). Therefore, in the following paragraphs, only the main effects of genotype and nutrient concentration will be presented, except for that single case.

*3.2. Pot Experiment*

3.2.1. Effect of Genotypes

Overall, 406B produced greater biomass than 405W, with significantly larger shoot DM (45%, 43%) and root DM (70%, 85%), in the P and K treatments of the experiment, respectively (Figure 1A,B,D,E). Likewise, 406B was also a taller than 405W (36%, 29%) in the P and K treatments, respectively (Tables 1 and 2). 406B flowered about 8 days later (Tables 1 and 2) and produced 45% less tillers than 405W (Tables 1 and 2). Differences in



panicle length and SPAD between the genotypes were less than 10% (Tables 1 and 2), and there was no difference in lodging index. In the K segment of the experiment, 406B had a significantly smaller HI (30% Table 2). Genotype had no significant effect on yield in the P segment of the experiment (Figure 1C). In the K segment of the experiment, at 40 and 80 mg L$^{-1}$ K, 406B had a larger yield, but at 10 and 20 mg L$^{-1}$ K, 406W had a larger yield (Figure 1F).

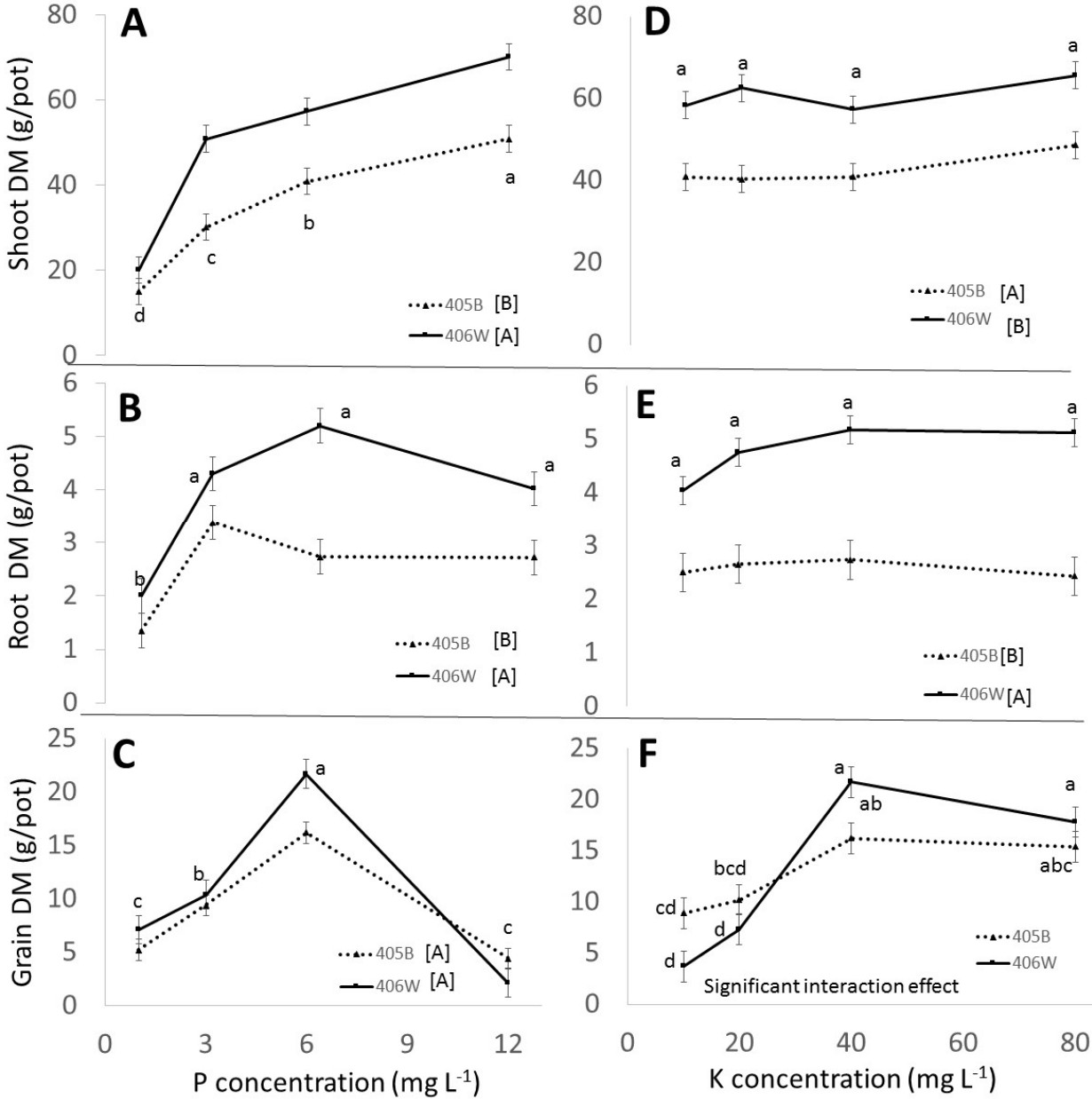

**Figure 1.** Shoot (**A**), root (**B**) and grain (**C**) dry matter at maturity in the winter pot experiment by P fertigation level, and Shoot (**D**), root (**E**) and grain (**F**) dry matter at maturity in the winter pot experiment by K fertigation level. Dashed line represents 405B genotype, and solid line represents 406W genotype. Different letters represent significant (<0.05) differences for main effects. Lower case letters adjacent to the trend line represent the results of the Tukey test for nutrient concentration averaged across genotypes, and capital letters adjacent to the legend represent the results of the Student-*t* test for the two different genotypes.

**Table 1.** Effect of P fertigation levels on growth parameters in the winter pot experiment. Different letters represent significant (<0.05) differences for main effects. Upper case letters next to the values in the 3rd row represent the results of the Tukey test for nutrient concentration averaged across genotypes, and lower case letters next to the values in the 4th row represent the results of the Student-*t* test for the two different genotypes.

| | Phosphorus Treatment (mg L$^{-1}$) | Height (cm) | Panicle Length (cm) | Days to Flowering | Harvest Index | SPAD | Number of Tillers per Plant | Lodging Index |
|---|---|---|---|---|---|---|---|---|
| | 1 | 35.5 | 32.8 | 50.6 | 0.30 | 41.6 | 5.3 | 1.2 |
| Genotype 405B | 3 | 41.6 | 39.4 | 47.6 | 0.24 | 40.0 | 10.7 | 1.1 |
| | 6 | 45.9 | 40.6 | 47.0 | 0.29 | 40.6 | 10.2 | 1.9 |
| | 12 | 45.4 | 43.2 | 45.0 | 0.08 | 40.0 | 16.6 | 1.7 |
| | 1 | 50.5 | 32.8 | 57.0 | 0.29 | 39.1 | 2.2 | 1.1 |
| Genotype 406W | 3 | 59.1 | 36.2 | 56.4 | 0.17 | 38.8 | 6.3 | 1.4 |
| | 6 | 60.0 | 37.8 | 53.8 | 0.28 | 38.0 | 5.5 | 1.9 |
| | 12 | 58.9 | 37.2 | 51.6 | 0.03 | 37.8 | 9.4 | 1.8 |
| | 1 | 43.0 B | 32.8 A | 53.8 A | 0.29 A | 40.3 A | 3.8 C | 1.1 B |
| Tukey for P Treatments | 3 | 50.3 A | 37.8 A | 52.0 AB | 0.21 A | 39.9 A | 8.5 B | 1.2 B |
| | 6 | 52.9 A | 39.2 A | 50.1 BC | 0.28 A | 39.3 A | 7.8 B | 1.9 A |
| | 12 | 52.1 A | 40.4 A | 48.3 C | 0.05 B | 38.8 A | 13.0 A | 1.7 A |
| Tukey for Genotype | 405B | 42.1 b | 39.0 a | 47.5 b | 0.23 a | 40.8 a | 10.7 a | 1.5 a |
| | 406W | 57.1 a | 36.0 b | 55.7 a | 0.19 a | 38.4 b | 5.8 b | 1.5 a |
| P x Genotype | N.S. | N.S. | N.S. | N.S. | N.S. | N.S. | N.S. | N.S. |

N.S. = Non-significant.

3.2.2. Effect of P

P fertilization had a significant effect on the DM of the shoot, root, and grain (Figure 1A–C) in both genotypes. The shoot DM was lowest at 1 mg L$^{-1}$ P (17.5 g), rose sharply until 3 mg L$^{-1}$ P and rose gradually thereafter until 12 mg L$^{-1}$ (Figure 1A). The root DM was lowest at 1 mg L$^{-1}$ P and rose sharply until 3 mg L$^{-1}$ P and did not rise significantly as the P rose over 3 mg L$^{-1}$ (Figure 1B). The grain DM was low at 1 mg L$^{-1}$ P, rose sharply until 6 mg L$^{-1}$ P, and declined drastically between 6 mg L$^{-1}$ P and 12 mg L$^{-1}$ P.

At flowering, P fertilization had a significant effect on shoot P concentration (Figure 2A) that rose by 400% between the 1 mg L$^{-1}$ P treatment and the 12 mg L$^{-1}$ P treatment. Also at maturity, P fertilization had a significant effect on shoot P concentration (Figure 2B). From 1 mg L$^{-1}$ P to 6 mg L$^{-1}$ P, there was no significant change in shoot P concentration, ranging between 0.04% and 0.09% (Figure 2A), but between 6 and 12 mg L$^{-1}$, the P concentration rose sharply to 0.37% (Figure 2A). The grain showed a steady increase between 0.28% and 0.54% P as the P fertilization was increased from 1 to 12 mg L$^{-1}$ P (Figure 2C).

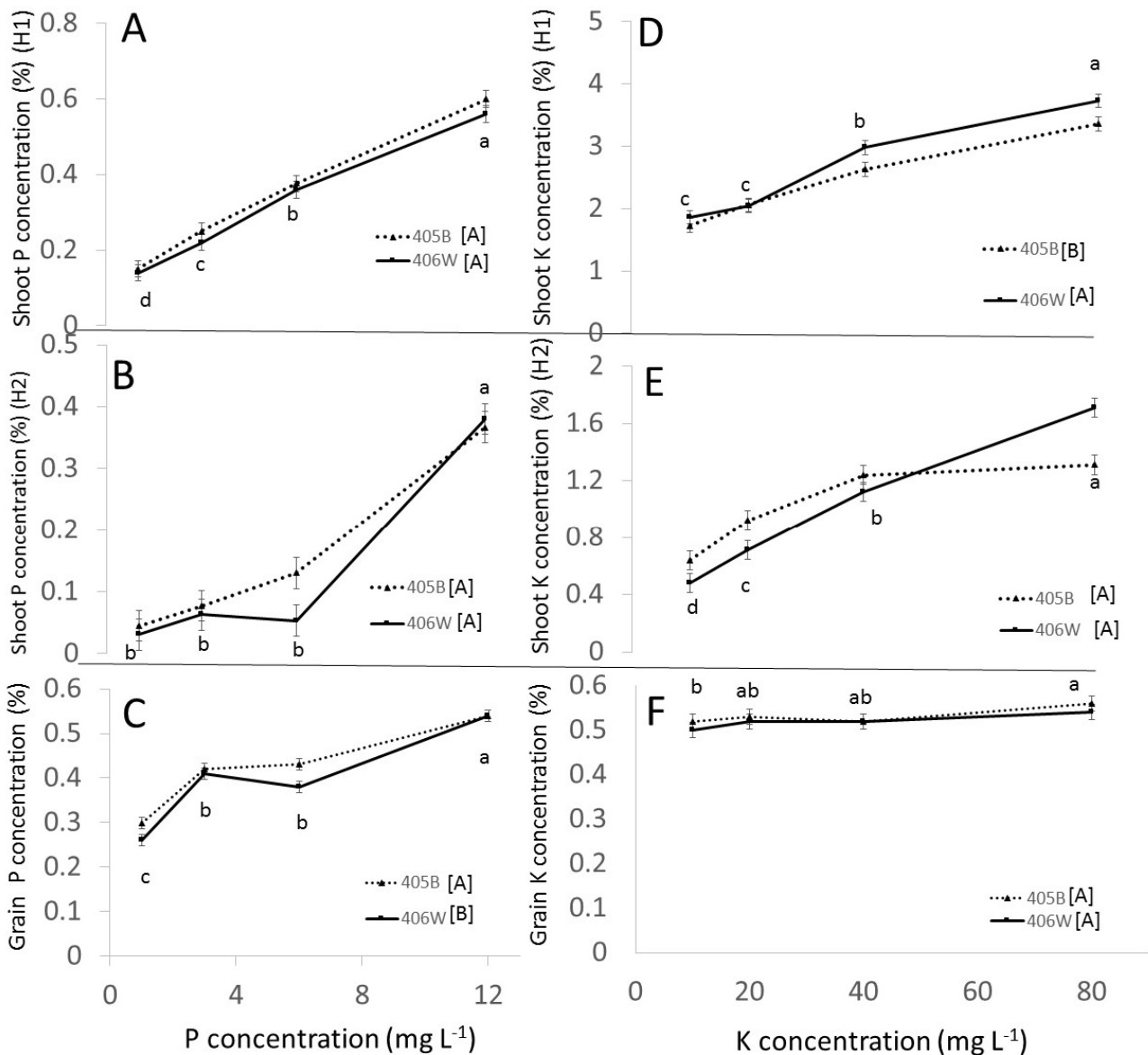

**Figure 2.** Shoot P concentration at flowering (**A**), shoot P concentration at maturity (**B**), and grain P concentration (**C**) in the winter pot experiment by P fertigation level. Shoot K concentration at flowering (**D**), shoot K concentration at maturity (**E**) and grain K concentration (**F**) in the winter pot experiment by K fertigation level. Dashed line represents 405B genotype, and solid line represents 406W genotype. Different letters represent significant (<0.05) differences for main effects. Lower case letters adjacent to the trend line represent the results of the Tukey test for nutrient concentration averaged across both genotypes, and capital letters adjacent to the legend represent the results of the Student-*t* test for the two different genotypes.

Several other growth parameters were affected by P fertilization (Table 1). Tef plants grown under the lowest P treatment (1 mg L$^{-1}$ P) were significantly shorter than the other treatments. The panicle length was unaffected by P fertilization, as was the SPAD. Lower P treatments took more time to reach 50% flowering than the higher P treatments, and the lodging index was higher in the high P treatments. Interestingly, the number of tillers increased as P fertilization was increased, and at the highest P treatment the HI fell dramatically.

### 3.2.3. Effect of K

K fertilization had no significant effect on shoot or root DM (Figure 1D,E). Grain DM was lowest at 10 mg L$^{-1}$ K and rose until 40 mg L$^{-1}$ K, with no significant changes between 40 mg L$^{-1}$ K and 80 mg L$^{-1}$ K (Figure 1F).

K fertilization had a significant effect on shoot K concentration at flowering (Figure 2D) and at maturity. At flowering, K concentration in the shoot rose steadily from 1.8% to 3.6% as the K treatment increased from 10 to 80 mg L$^{-1}$. At maturity, K concentration in the shoot rose from 0.6% to 1.5% as the K treatments increased from 10 to 80 mg L$^{-1}$ (Figure 2E). However, the effect of K fertilization on grain K concentration was negligible (albeit statistically significant) between the highest and lowest treatments, with the lowest being 0.48% and the highest being 0.55% (Figure 2F).

Other growth parameters were also affected by K fertilization (Table 2). The plants were 11% taller in the highest K treatment compared to the lowest K treatment, and the plants took 4 days more to reach 50% flowering in the lowest K treatment compared to the highest one. The lower K treatments had more tillers and less lodging, and the higher K treatments had a higher HI.

**Table 2.** Effect of K fertigation levels on growth parameters in the winter pot experiment. Different letters represent significant (<0.05) differences for main effects. Upper case letters next to the values in the 3rd row represent the results of the Tukey test for nutrient concentration averaged across genotypes, and lower case letters next to the values in the 4th row represent the results of the Student-*t* test for the two different genotypes.

|  | Potassium Treatment (mg L$^{-1}$) | Height (cm) | Panicle Length (cm) | Days to Flowering | Harvest Index | SPAD | Number of Tillers per Plant | Lodging Index |
|---|---|---|---|---|---|---|---|---|
| Genotype 405B | 10 | 42.1 | 37.0 | 49.2 | 0.18 | 41.0 | 14.2 | 1.8 |
|  | 20 | 45.6 | 40.8 | 47.6 | 0.20 | 41.5 | 13.5 | 1.9 |
|  | 40 | 45.8 | 40.6 | 47.0 | 0.29 | 40.6 | 10.2 | 1.9 |
|  | 80 | 46.7 | 37.2 | 45.6 | 0.24 | 40.8 | 10.0 | 2.5 |
| Genotype 406W | 10 | 54.0 | 38.0 | 57.8 | 0.06 | 39.2 | 7.3 | 1.6 |
|  | 20 | 56.4 | 37.2 | 56.4 | 0.11 | 38.9 | 8.0 | 1.7 |
|  | 40 | 56.7 | 37.8 | 53.8 | 0.28 | 38.0 | 5.5 | 1.9 |
|  | 80 | 61.8 | 40.2 | 53.8 | 0.21 | 38.0 | 5.8 | 2.7 |
| Tukey for K Treatments | 10 | 48.0 B | 37.5 A | 53.0 A | 0.12 C | 40.1 A | 10.8 A | 1.7 B |
|  | 20 | 51.0AB | 39.0 A | 52.0 AB | 0.16 BC | 40.2 A | 10.7 A | 1.8 B |
|  | 40 | 52.9 A | 39.2 A | 50.4 B | 0.28 A | 39.3 A | 7.8 B | 1.9 AB |
|  | 80 | 54.2 A | 40.9 A | 49.7 B | 0.22 AB | 39.4 A | 7.8 B | 2.6 A |
| Tukey for Genotype | 405B | 45.1 b | 40.0 a | 47.3 b | 0.23 a | 41.0 a | 12.0 a | 2.0 a |
|  | 406W | 58.0 a | 38.0 a | 55.4 a | 0.16 b | 38.6 b | 6.6 b | 2.0 a |
| K x Genotype | N.S. | N.S. | N.S. | N.S. | N.S. | N.S. | N.S. | N.S. |

N.S. = Non-significant.

### 3.3. *Field Experiment*

#### 3.3.1. Effect of Genotypes

In the field, both cultivars had relatively low yields and low HI regardless of the P and K treatments (Figure 3B,D, Tables 3 and 4). It is worth noting that 406B had a significantly smaller grain yield than 405W. Overall, the two cultivars responded to P and K treatments in a similar way (Figures 3 and 4, Tables 3 and 4).

3.3.2. Effect of P and K

At maturity, the shoot DM was larger in the treatments that received P compared to the 0 P treatment, but there was no significant difference in shoot DM between any of the P treatments above 0 (Figure 3B). At flowering, shoot DM follows the same trend but the there are no statistically significant differences (Figure 3A). The grain yield was not significantly affected by P treatment in the field, but there was a trend towards greater grain yield as the P concentration in the irrigation water rose from 0 to 6 mg $L^{-1}$, and a moderate, statistically insignificant decline thereafter (Figure 3C).

At maturity, the P concentration in the shoot was larger in the treatments that received P compared to the 0 P treatment, but there was no significant difference in the shoot P concentration between any of the P treatments above 0 (Figure 4B). At flowering, P concentration in the shoot showed no clear trend. The P treatments had no significant effect on grain P concentration (Figure 4C).

At flowering, there was no effect of K fertilization on shoot dry weight (Figure 3D). However, at maturity, the shoot dry weight and the grain yield responded favorably to K fertilization. Shoot DM increased by 27% and grain DM and 35%, respectively, between the 0 and 80 mg $L^{-1}$ K treatments (Figure 3E,F). At flowering, K concentration in the shoots was not significantly affected by K fertilization (Figure 4D). At maturity, the K concentration in the shoots rose by 12% between the 0 and 80 mg $L^{-1}$ K treatments (Figure 4E). K concentration in the grain was unaffected by K treatments (Figure 4D).

None of the other growth parameters (height, panicle length, days to flowering, harvest index, SPAD, or number of tillers) were significantly affected by P or K fertilization levels in the field experiment (Tables 3 and 4).

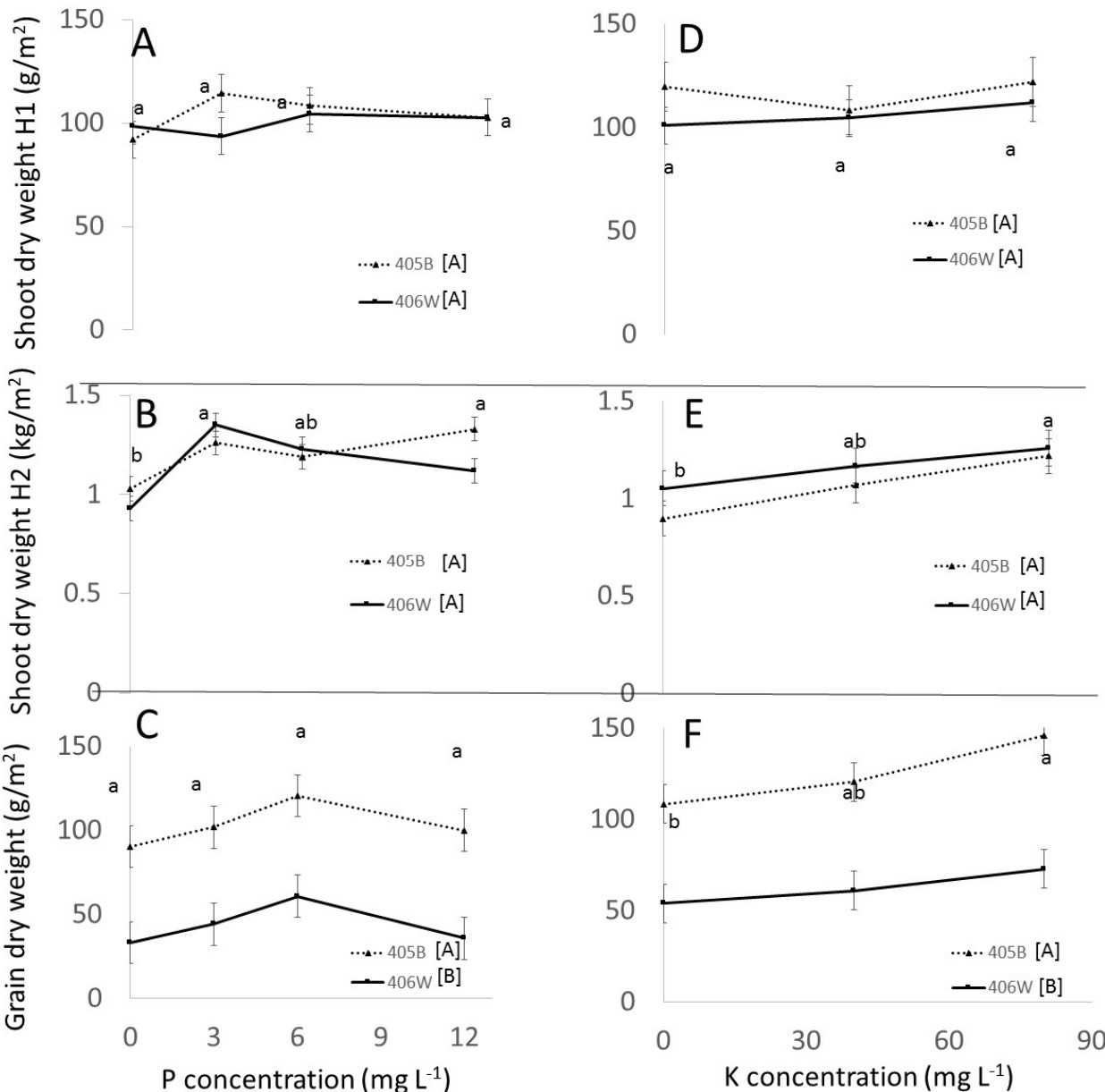

**Figure 3.** Shoot dry matter at flowering (**A**), Shoot dry matter at maturity (**B**) and grain dry matter (**C**) in the summer field experiment by P fertigation level, and Shoot (**D**), root (**E**) and grain (**F**) dry matter in the summer field experiment by K fertigation level. Dashed line represents 405B genotype, and solid line represents 406W genotype. Different letters represent significant (<0.05) differences for main effects. Lower case letters adjacent to the trend line represent the results of the Tukey test for N concentration averaged across genotypes, and capital letters adjacent to the legend represent the results of the Student-*t* test for the two different genotypes.

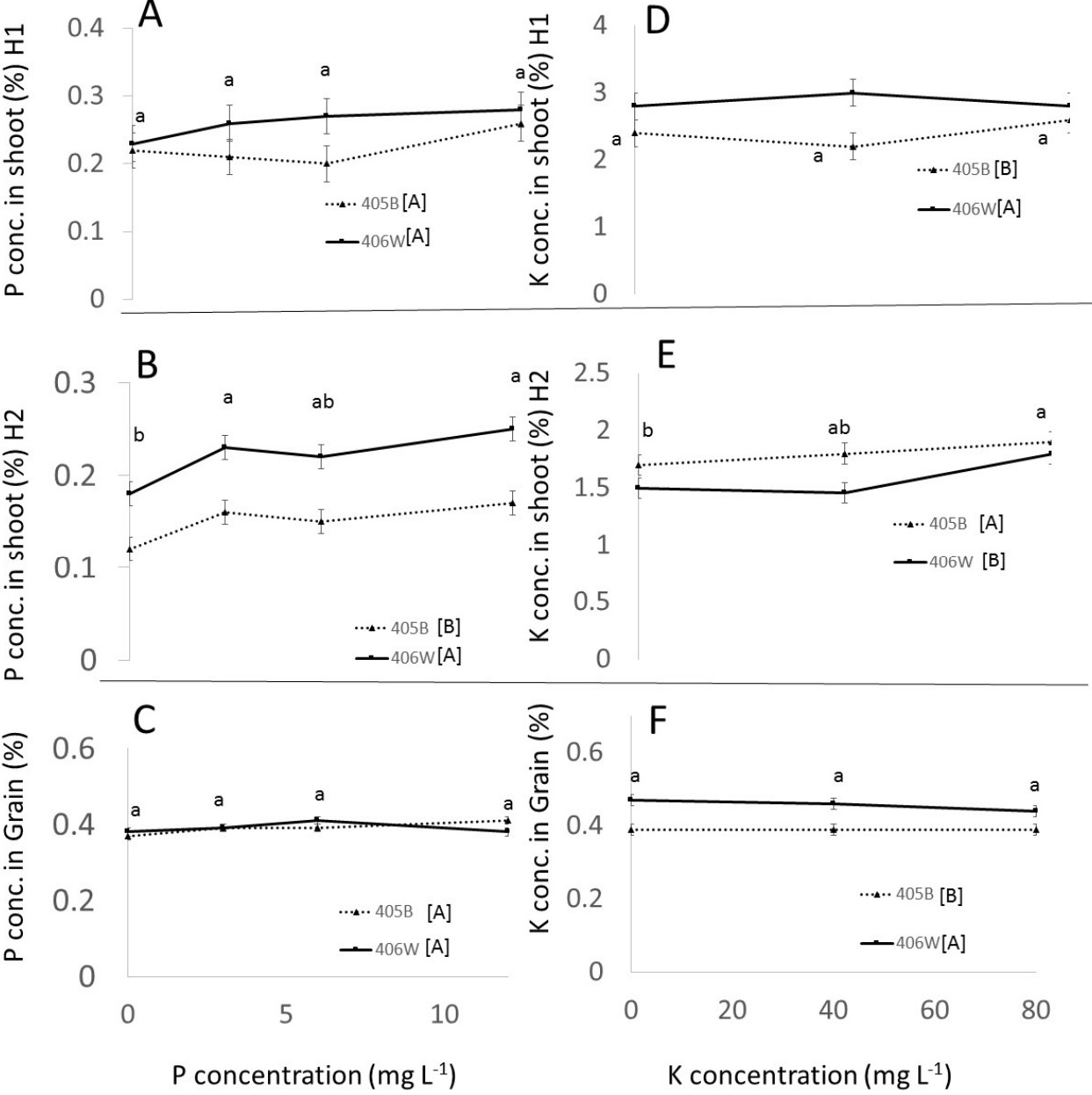

**Figure 4.** Shoot P concentration at flowering (**A**), shoot P concentration at maturity (**B**), and grain P concentration (**C**) in the summer field experiment by P fertigation level. Shoot K concentration at flowering (**D**), shoot K concentration at maturity (**E**) and grain K concentration (**F**) in the summer field experiment by K fertigation level. Dashed line represents 405B genotype, and solid line represents 406W genotype. Different letters represent significant (<0.05) differences for main effects. Lower case letters adjacent to the trend line represent the results of the Tukey test for N concentration averaged across genotypes, and capital letters adjacent to the legend represent the results of the Student-*t* test for the two different genotypes.

**Table 3.** Effect of P fertigation levels on growth parameters in the summer field experiment. Different letters represent significant (<0.05) differences for main effects. Upper case letters next to the values in the 3rd row represent the results of the Tukey test for nutrient concentration averaged across genotypes, and lower case letters next to the values in the 4th row represent the results of the Student-*t* test for the two different genotypes.

| | Phosphorus Treatment (mg L⁻¹) | Height (cm) | Panicle Length (cm) | Days to Flowering (50%) | Harvest Index | SPAD | Number of Tillers per Plant |
|---|---|---|---|---|---|---|---|
| | 1 | 95.6 | 32.2 | 40.4 | 0.09 | 35.4 | 14.6 |
| Genotype 405B | 3 | 95.9 | 31.5 | 37.8 | 0.08 | 33.5 | 17.2 |
| | 6 | 100.7 | 33.1 | 37.4 | 0.10 | 37.4 | 13.8 |
| | 12 | 102.3 | 32.0 | 37.0 | 0.07 | 34.9 | 19.2 |
| | 1 | 94.7 | 24.8 | 54.2 | 0.04 | 36.8 | 8.8 |
| Genotype 406W | 3 | 97.3 | 30.7 | 54.2 | 0.03 | 34.9 | 12.8 |
| | 6 | 94.6 | 34.7 | 56.4 | 0.05 | 37.4 | 11.2 |
| | 12 | 97.7 | 34.6 | 56.4 | 0.03 | 37.0 | 10.8 |
| | 1 | 95.6 A | 28.5 A | 47.3 A | 0.06 A | 36.1 A | 11.7 A |
| Tukey for P Treatments | 3 | 97.3 A | 31.1 A | 46.0 A | 0.06 A | 34.2 A | 15.0 A |
| | 6 | 94.6 A | 33.9 A | 46.9 A | 0.08 A | 37.3 A | 12.5 A |
| | 12 | 97.7 A | 33.3 A | 46.7 A | 0.05 A | 37.0 | 15.0 A |
| Tukey for Genotype | 405B | 98.9 a | 32.2 a | 38.2 b | 0.09 a | 35.3 a | 16.2 a |
| | 406W | 96.1 a | 31.2 a | 55.3 a | 0.04 b | 36.5 a | 10.9 a |
| P x Genotype | N.S. | N.S. | N.S. | N.S. | N.S. | N.S. | N.S. |

N.S. = Non-significant.

**Table 4.** Effect of K fertigation levels on growth parameters in the summer field experiment. Different letters represent significant (<0.05) differences for main effects. Upper case letters next to the values in the 3rd row represent the results of the Tukey test for nutrient concentration averaged across genotypes, and lower case letters next to the values in the 4th row represent the results of the Student-*t* test for the two different genotypes.

| | Potassium Treatment (mg L⁻¹) | Height (cm) | Panicle Length (cm) | Days to Flowering (50%) | Harvest Index | SPAD | Number of Tillers per Plant |
|---|---|---|---|---|---|---|---|
| | 0 | 101.0 | 31.8 | 37.8 | 0.11 | 35.0 | 17.0 |
| Genotype 405B | 40 | 100.7 | 33.1 | 37.4 | 0.10 | 34.4 | 13.8 |
| | 80 | 92.6 | 26.6 | 38.6 | 0.11 | 37.2 | 20.8 |
| | 0 | 108.0 | 32.5 | 60.0 | 0.05 | 37.1 | 11.0 |
| Genotype 406W | 40 | 94.6 | 34.6 | 56.4 | 0.05 | 37.3 | 11.2 |
| | 80 | 97.2 | 32.7 | 55.0 | 0.06 | 37.3 | 10.2 |
| | 0 | 104.5 A | 32.1 A | 48.9 A | 0.08 A | 36.0 A | 14.0 A |
| Tukey for K Treatments | 40 | 97.6 A | 33.9 A | 46.9 A | 0.08 A | 37.4 A | 12.5A |
| | 80 | 94.9 A | 29.7 A | 46.8 A | 0.08 A | 37.3 A | 15.5 A |
| Tukey for Genotype | 405B | 98.1a | 30.5a | 37.9 b | 0.11 a | 36.5 a | 17.2 a |
| | 406W | 99.9 a | 33.3a | 57.1 a | 0.05 b | 37.2 a | 10.8 b |
| K x Genotype | N.S. | N.S. | N.S. | N.S. | N.S. | N.S. | N.S. |

N.S. = Non-significant.

## 4. Discussion

Similar to the response of tef to N fertilization detailed by Gashu et al. [4], the response to P in the pot experiment can be divided into three ranges: underfertilization, sufficient fertilization, and overfertilization. As fertilization level rose from 1 mg L⁻¹ P to 6 mg L⁻¹ P, the grain yield increased by over threefold, indicating that section of the graph was underfertilized (Figure 1C). At 12 mg L⁻¹, the yield decreased dramatically, indicating overfertilization. The optimal P concentration is, therefore, between 6 and 12 mg L⁻¹, and

future experiments should include treatments within this range for a clearer picture. Interestingly, while overfertilization severely impacted the grain yield, it did not have any negative effect on the shoot or root DM (Figure 1A,B).

At flowering, P concentration in the shoot increased linearly with added P regardless of the concentration (Figure 2A). However, at maturity, between 1 mg $L^{-1}$ P and 6 mg $L^{-1}$ P, there was no significant increase in shoot P, while between 6 and 12 mg $L^{-1}$ P the shoot P concentration increased by 400% (Figure 2B). Hawkesford et al. [17] stated that plants rarely show signs of P toxicity since they are able to downregulate P transporters involved in net root P uptake. This seems to be the case between 1 and 6 mg $L^{-1}$ P, since at maturity (Figure 2B) the concentration in the shoot was not significantly increased in that range. The sharp increase in shoot P concentration between 6 and 12 mg $L^{-1}$ seems to indicate that the tef plant was unable to regulate the P concentration effectively. That being said, there was little evidence of actual toxicity, since other than a reduction in grain yield, plant growth was unaffected by P overfertilization (Figure 1A,B). Given the large number of tillers in the tef that received 12 mg $L^{-1}$ P compared to the other treatments (Table 1), it seems that excess P simply caused an increase in tiller production at the expense of grain production. The detrimental effects of P overfertilization were reported by Girma et al. [18] in a field trial in Oklahoma, who found that even low levels of P fertilization decreased yield.

In the field experiment, there was a trend towards higher yields at 6 mg $L^{-1}$ P, which is similar to the pot experiment, but the difference in grain yield between the minimum and maximum was 33% but statistically insignificant (Figure 3B). It is noteworthy that the grain yield and HI in the field were exceedingly low. Possible reasons for this have been discussed in detail by Gashu et al. [4], including the heat and long daylight hours in mid-summer months.

It is instructive to compare the P concentration in the shoot in the field to that in the pot experiment at flowering in order to determine, at least on a basic level, what degree of P availability occurred in the field under different treatments. In the field, shoot P concentration at flowering under 0 P application, averaged across cultivars, was 0.22% (Figure 4A), while under the 12 mg $L^{-1}$ P treatment it was 0.27%. Both these concentrations are close to the P concentration found in the pot experiment in the shoot at flowering time (0.24%) for the 3 mg $L^{-1}$ P treatment (Figure 2A). This suggests that regardless of the treatment, the P availability in the soil was approximately 3 mg $L^{-1}$ P in the root zone. Evidently, there was already some available P in the soil, somewhat below the 6 mg $L^{-1}$, which gave an optimum grain yield in the pot experiment. However, adding P did not increase the availability of P in the field at flowering. Perhaps the added P was immobilized through biotic or abiotic processes.

The P concentration at maturity tells a slightly different story. At maturity, plots receiving 0 mg $L^{-1}$ in the irrigation water had a significantly smaller P concentration in the shoots than those that received even low amounts of P concentrations (Figure 4B). Furthermore, the total P uptake was significantly higher in those plots that received P in the irrigation water, with no significant difference between those treatments (data not shown). Evidently, as the plant grew larger and P became more limiting, there was a significant advantage to fertigating with P.

This may explain why the grain yield at 6 mg $L^{-1}$ in the field was 33% higher than the grain yield when 0 P was applied, though not a statistically significant difference. This follows the same trend as the pot experiment, in which grain yield reached a maximum at 6 mg $L^{-1}$ P.

In the pot experiment, the response to K fertilization can be divided into two ranges: underfertilization and sufficient fertilization. The grain DM responded positively to K fertilization up to 40 mg $L^{-1}$ K, and then plateaued (Figure 1F). Unlike the P (Figure 1A–C) and N (Gashu et al., 2020) segments of this experiment, there was no significant negative effect of K at the highest level of K fertilization, although there was a small, non-significant decline in grain DM between the 40 mg $L^{-1}$ and 80 mg $L^{-1}$ treatment (Figure 1F). This is

similar to the response of tef grain yield to K fertilization observed by Misskire et al. [19] in a field experiment in Ethiopia. They reported yield rising and then plateauing with increased K, with a slight, non-significant drop at the highest K level. It is interesting that no decline in grain yield occurred in the higher K treatments, since Mulugeta et al. [9] found a decline in grain yield in plants with much lower shoot K concentrations (0.6–0.9%) than the concentrations we measured in the highest K fertilization treatments (Figure 2C). Furthermore, maximum yield was observed at a much higher vegetative K concentration in our experiment (1%) compared to the 0.63% K reported by Mulugeta et al. [9] as the internal K requirement of tef. It is possible that the decreased grain yield in high K treatments reported by Mulugeta et al. [9] was not caused by the increase in K itself, but rather a secondary effect caused by an imbalance of other nutrients. In our experiment, all other nutrients were provided at high rates and in available forms, so imbalances of other nutrients did not occur. This underscores the importance of conducting experiments in perlite, where minerals can be provided in a highly controlled manner.

Interestingly, in the pot experiment, there were no significant changes in shoot or root biomass across the entire K treatment range (Figure 1D,E). In the field experiment there was an increase in shoot DM as the K concentration in the fertigation water increased from 0 to 80 mg $L^{-1}$, but this difference was not apparent at flowering and became apparent only at maturity (Figure 2E). The lack of response of the vegetative portions of the plant to K fertilization in the pot experiment and the late-stage response in the field experiment shows that the completion of the lifecycle of the plant and measurement of grain yield are essential for showing the effect of K fertilization on tef. The increase in straw yield due to K fertilization which we observed in the field and not in the pot experiment has been reported previously. Gebrehawariat et al. [11] reported an increase in straw with increased K fertilization in tef. Interestingly, while Gebrehawatiat et al. [11] reported that increased K caused an increase in the number of fertile tillers, we observed a decrease in tillers with increased K in the pot experiment (Table 2), and no clear effect of K on tiller number in the field experiment (Table 4).

In the pot experiment, K concentration in the shoot increased with increasing K fertilization (Figure 2D,E), but the K concentration in the grain was not responsive to K fertilization, remaining at around 0.5% regardless of cultivar or K treatment. In the field experiment, there were significant differences between the cultivars (0.46% in 406W and 0.39% in 405B) but no significant effect of K treatment on grain K concentration. Evidently, the K concentration in the shoot is flexible, but the K concentration in the grain needed to be at a certain range in order to allow grain development. This is in contrast to the findings of Gebrehawariyat et al. [11], who found that on average grain K concentration increased by 33% from the lowest K treatment (0 kg $K_2O$ ha$^{-1}$) to the highest K treatment (120 kg $K_2O$ ha$^{-1}$).

In the field, shoot K concentration at flowering under 0 K application, averaged across cultivars, was 2.6%, while under 80 mg $L^{-1}$ K treatment it was 2.7%. Both these concentrations are close to the K concentration found in the pot experiment in shoot at flowering (2.8%) under the 40 mg $L^{-1}$ K treatment (Figure 2D). This suggests that at the beginning of the field experiment, regardless of the treatment, the K availability in the soil was approximately 40 mg $L^{-1}$ K in the root zone. At maturity, however, there was a 16% increase in shoot K concentration between the 0 and 80 mg $L^{-1}$ K treatments, as well as a 56% increase in total K uptake between those two treatments (data not shown), indicating that fertigation with K increased K availability later on in the lifecycle.

## 5. Conclusions

The purpose of this work was to show the effects of P and K availability on tef growth variables. While the pot experiment showed these effects over a wide range of P and K availabilities, the field experiment demonstrated the effects over a much smaller range, evidently due to minerals available in the soil and interaction of applied nutrients with the soil. The main lessons that can be drawn from these experiments are as follows:

1. Up to 6 mg L$^{-1}$ P and 40 mg L$^{-1}$ K, respectively, there was a clear positive effect of P and K fertilization in the pot experiment. P and K concentration in the shoot was positively affected by P and K fertilization. The concentration of P in the grain was positively affected by P fertilization, whereas the K concentration in the grain was barely affected by K fertilization. These observations were true for both genotypes.
2. A clear negative effect of P overfertilization on grain yield was evident when perlite pots were fertigated with 12 mg L$^{-1}$ P. This effect could not be seen in the field and to the best of our knowledge has not been reported before. It seems to have been caused by overinvestment in tillers and underinvestment in grain. P fertilization tended to have a positive effect on the number of tillers.
3. We observed that the benefit of K fertilization was only evident at the end of the plants' lifecycle, which has practical ramifications for experimental design when testing the effect of K on tef.
4. No statistically significant negative effect from K overfertilization was observed in the pot or field experiment, even though the K concentration of the shoots at maturity was well above what was considered optimum in the literature.
5. While K concentration in the shoot was clearly affected by K availability, the K concentration in the grain remained more or less constant, around 0.5% regardless of K availability.
6. The response of tef to different P and K doses in the field was attenuated compared to the response in perlite, evidently because of the native ability of the soil to release and fix P and K. Fertigation recommendations in the field will need to take into account the nutrient availability in the soils.

These lessons, as well as quantitative data presented in this paper is, useful in understanding how tef responds to P and K fertilization. Together with the description of tef response to N presented in Gashu et al. (2020), this paper contributes to our understanding of how tef might be grown intensively both inside and outside Ethiopia.

**Author Contributions:** Conceptualization, K.G. and U.Y.; Data curation, K.G.; Formal analysis, M.H. and K.G.; Investigation, K.G., I.Z. and Y.S.; Methodology, I.Z., Y.S. and U.Y.; Project administration, U.Y.; Supervision, I.Z., Y.S. and U.Y.; Writing—original draft, M.H.; Writing—review and editing, M.H., K.G., Y.S. and U.Y. All authors have read and agreed to the published version of the manuscript.

**Funding:** This project was financed by the center of fertilizer and plant nutrition (CFPN).

**Acknowledgments:** We would like to thank Luda, Ina Finegold, Yonatan Saroya, Jajaw Bimro, Behailu Kassahun and Fanosie Mekonen for assistance. Y.S. is the incumbent of the Haim Gvati Chair in Agriculture.

**Conflicts of Interest:** The authors declare no conflict of interest.

## Appendix A

**Table A1.** List of treatments in winter pot experiment, including N treatments reported by Gashu et al. 2020.

| Segment | Mineral Concentrations in Irrigation Solution (ppm) | | | # of Genotypes | Repetitions |
|---|---|---|---|---|---|
| | Nitrogen (N) | Phosphorus (P) | Potassium (K) | | |
| Nitrogen Segment | 10 | 6 | 40 | 2 | 5 |
| | 20 | 6 | 40 | 2 | 5 |
| | 80 | 6 | 40 | 2 | 5 |
| | 120 | 6 | 40 | 2 | 5 |
| Phosphorus Segment | 40 | 1 | 40 | 2 | 5 |
| | 40 | 3 | 40 | 2 | 5 |
| | 40 | 12 | 40 | 2 | 5 |
| Potassium Segment | 40 | 6 | 10 | 2 | 5 |

| | 40 | 6 | 20 | 2 | 5 |
| --- | --- | --- | --- | --- | --- |
| | 40 | 6 | 80 | 2 | 5 |
| Shared control group | 40 | 6 | 40 | 2 | 5 |

**Table A2.** List of treatments in summer field experiment, including N treatments reported by Gashu et al. 2020.

| Segment | Mineral Concentrations in Irrigation Solution (ppm) | | | # of Genotypes | Repetitions |
| --- | --- | --- | --- | --- | --- |
| | Nitrogen (N) | Phosphorus (P) | Potassium (K) | | |
| Nitrogen Segment | **0** | 6 | 40 | 2 | 5 |
| | 30 | 6 | 40 | 2 | 5 |
| | 120 | 6 | 40 | 2 | 5 |
| Phosphorus Segment | 60 | 0 | 40 | 2 | 5 |
| | 60 | 3 | 40 | 2 | 5 |
| | 60 | 12 | 40 | 2 | 5 |
| Potassium Segment | 60 | 6 | 0 | 2 | 5 |
| | 60 | 6 | 80 | 2 | 5 |
| Shared Control group | 60 | 6 | 40 | 2 | 5 |

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
