# Peer review of "Tef (Eragrostis tef) Responses to Phosphorus and Potassium Fertigation under Semi-Arid Mediterranean Climate"

_agronomy, doi:10.3390/agronomy11081588_

Round 1
Reviewer 1 Report
The manuscript “Agronomy-1295294” entitled “Tef (Eragrostis tef) Responses to Phosphorus and Potassium Fertigation under Semi-Arid Mediterranean Climate” by Halpern et al. deals with an interesting subject regarding the responses of two tef genotypes to escalating phosphorus (P) and potassium (K) levels in terms of nutrient uptake, shoot, root, and grain biomass production, as well as growth parameters such as height, number of tillers, time to flowering, Harvest index, and SPAD. This study was carried out in two phases. In the first phase, tef was grown under controlled conditions in a greenhouse, in perlite, an inert medium that allowed full control of the nutrient concentration in the root zone. The second phase involved growing tef out in the field in a semi-arid Mediterranean climate in natural soil. While the first phase was designed to clarify the basic response of tef to different P and K concentrations, the second phase was intended to show how the lessons learned from the first phase plays out in a realistic agricultural environment. The same experiments were used to study N fertigation in tef, recently published by Gashu et al. (2020) at the same journal.
For publication in “Agronomy” journal, the topic and content are appropriate. Scientific content and the manuscript size are appropriate. The introduction provides sufficient background and includes all relevant references. The novelty of the results is above the average of novelty knowledge. The discussion is sufficient. The conclusions of the article are well-proof. The title and the abstract of the manuscript are also appropriate. The editing and linguistic quality are good. When figures and tables are shown, the units are correctly used. The quality of citations is appropriate. In general, the quality of the experiment is well performed and follows rigid scientific logic. However, there are some points that need attention in order for the article to be published:
- The abstract is too long (408 words when the limit is about 200 words) and descriptive.
- The affiliation section must be revised according to “Instructions for authors”.
- Keywords: Please change some keywords. Title and keywords must not contain the same words.
- Authors should rewrite the first two paragraphs of introduction section as these paragraphs seem to be the same with those of their previous research article studied N fertigation in tef (Gashu et al., 2020).
- Line 60: “30-40 kg P ha-1” instead of “30-40 kg ha-1 P”
- Lines 119-120: “mg L-1” instead of “ppm”
- Line 177: Please add more information about the determination of K concentration and add a reference.
- Line 193: Add information about the manufacturer of statistical software.
- Line 196: “Main and Interactive Effects” should be included as a subsection in the “Pot Experiment” section.
- Line 245: Please complete the sentence.
- Tables 1-4: Please explain what is N.S. and #
- Figures and Tables: “mg L-1” instead of “ppm”
- The discussion section must be enhanced. The authors should further refer to previous studies with regard to phosphorus and potassium fertilization. For instance, they can include the following papers in order to enrich the discussion section:
- Misskire, Y.; Mamo, T.; Taddesse, A.M.; YermiyahuThe effect of potassium on yield, nutrient uptake and efficiency of teff (Eragrostis tefZucc. Trotter) on vertisols of North Western Ethiopian Highlands. J. Plant Nutr., 2019, 42, 307-322, DOI: 10.1080/01904167.2018.1554681
- Girma, K.; Reinert, M.; Ali, M.S.; Sutradhar, A.; Mosali, J. 2012. Nitrogen and phosphorus requirements of teff grown under dryland production system. Crop Management, 2012, 11, 1-14. DOI:10.1094/CM-2012-0319-02-RS.
- Lines 502-504: The correct is: “Roussis, I.; Folina, A.; Kakabouki, I.; Kouneli, V.; Karidogianni, S.; Chroni, M.; Bilalis, D. Effect of organic and inorganic fertilization on yield and yield components of teff [Eragrostis tef(Zucc.) Trotter] cultivated under Mediterranean semi-arid conditions. Pap. Ser. A Agron. 2019, 62, 138–144.”
- References: The scientific name “Eragrostis tef” must be in italics. Please check all the references.
- Finally, please check and improve the Tables and Figures according to journal style.
Thank you for your consideration.
Author Response
I have attached a file with my response

Reviewer 2 Report
This MS tried to evaluate the effects of P and K application on Tef cropping system. There are some interesting results on Tef growth under different P and K treatments. But the current MS need more improvements.
Abstract:
Lines11-15, this sentence is too long and it is difficult to understand.
Line 15-21, The method introduction is too in detail. PLS summarize it.
In the abstract, PLS show the significant results and your new findings.
Introduction,
The scientific problems and objectives are not clear.
Lines 86-95, This is method, Pls put it into section of materials and methods.
Materials and methods
Line 101-102, It is meaningless on N.
For the experiment, it is too hard to read and follow so many introductions on method and data collections. Pls add tables.
Please add the measurement method on soil and crop P and K contents.
Results,
PLS focuses on the title and rewrite it to show how the effect of P on Tef growth, content, different genotypes, and how the effect of K application on Tef growth, content, different genotypes.
Discussion,
The current discussion is too tedious long and did not make clear research ideas. In general, it needs to discuss based on the objectives.
Conclusion,
Also, it is too long.
PLS rewrite it and figure out the major findings.
Author Response
I have attached a file with my response